# Gender Differences in Cortisol and Cortisol Receptors in Depression: A Narrative Review

**DOI:** 10.3390/ijms24087129

**Published:** 2023-04-12

**Authors:** Chuin Hau Teo, Ally Chai Hui Wong, Rooba Nair Sivakumaran, Ishwar Parhar, Tomoko Soga

**Affiliations:** Jeffrey Cheah School of Medicine and Health Sciences, Monash University Malaysia, Bandar Sunway, Kuala Lumpur 47500, Selangor, Malaysia

**Keywords:** stress, gender, depression, cortisol, glucocorticoid receptors, mineralocorticoid receptors

## Abstract

Stress is known to have a significant impact on mental health. While gender differences can be found in stress response and mental disorders, there are limited studies on the neuronal mechanisms of gender differences in mental health. Here, we discuss gender and cortisol in depression as presented by recent clinical studies, as well as gender differences in the role of glucocorticoid receptors (GRs) and mineralocorticoid receptors (MRs) in stress-associated mental disorders. When examining clinical studies drawn from PubMed/MEDLINE (National Library of Medicine) and EMBASE, salivary cortisol generally showed no gender correlation. However, young males were reported to show heightened cortisol reactivity compared to females of similar age in depression. Pubertal hormones, age, early life stressors, and types of bio-samples for cortisol measurement affected the recorded cortisol levels. The role of GRs and MRs in the HPA axis could be different between males and females during depression, with increased HPA activity and upregulated MR expression in male mice, while the inverse happened in female mice. The functional heterogeneity and imbalance of GRs and MRs in the brain may explain gender differences in mental disorders. This knowledge and understanding will support the development of gender-specific diagnostic markers involving GRs and MRs in depression.

## 1. Introduction

Mental health is an important part of a person’s well-being and, in many parts of the world, is considered one of the top priorities in public health today. Ever since the COVID-19 outbreak resulted in a global pandemic in March 2020 [1], movement restrictions have had a large negative impact on the mental health of the general population worldwide. Measures taken by health authorities to reduce the spread of the disease, such as lockdowns, have also contributed to economic recession, which is another factor in the increase in psychological distress, anxiety, stress, and panic [2,3,4,5]. As such, the importance of mental health, as well as mental health care, particularly in the field of anxiety and depressive symptoms, is now more significant than ever.

One of the important biomarkers in clinical studies for stress and depression is cortisol, a glucocorticoid hormone regulated by the hypothalamic-pituitary-adrenal (HPA) axis that is postulated to play an important role in mental health outcomes, particularly depression [6]. In a healthy and stress-free individual, cortisol is released according to the circadian rhythm and can be divided into two phases: the cortisol awakening response (CAR) and the diurnal cortisol slope (DCS). The first phase, CAR, involves a rapid increase in cortisol levels upon a person waking up, generally reaching peak levels after 30 to 45 min. The second phase, DCS, follows with a gradual decrease in cortisol levels throughout the rest of the day [7]. Any imbalances or changes in the pattern of cortisol release can become a contributing factor to negative mental health outcomes including depression.

Though depression is a complex disease that can have many causes, females are known to be at higher risk of being diagnosed with depression compared to males, which is a phenomenon that could be attributed to the exposure of chronic stressors as well as the effect of gonadal hormones [8]. Females experience more hormonal changes during their lifetime through puberty, menstruation, pregnancy, and menopause, which could make them more susceptible to mental illnesses such as depression [9]. However, while there has been research regarding gender differences in stress and depression, few studies have set out to specifically investigate the difference in the release of cortisol between females and males in relation to depression.

Glucocorticoids have been known to play an important role in mental health for a long time. Released during stress, glucocorticoids have been associated with depression and mood disorders [10]. It stands to reason that the receptors for these glucocorticoids are also important subjects of study. Glucocorticoid receptors (GRs) and mineralocorticoid receptors (MRs) are the target receptors in the hypothalamic-pituitary-adrenal (HPA) axis during stress [11]. Increasing evidence suggests that GRs and MRs have different affinities and binding activities to cortisol [12]. Both receptors show different reactions to cortisol and functionally interact with each other during stress in the HPA axis seen in clinical and animal studies [12].

While changes in GR and MR functions in association with stress are known to be associated with mental health disorders, more recent research also includes gender differences in the HPA’s response to stress [13]. GRs and MRs have different effects on mood control and the related behavior in males and females [14]. What works as a treatment for one gender may not be as effective for the other. There is also increasing evidence that GRs and MRs play a key role in the functionality of the HPA axis and determine susceptibility to stress-related mental disorders [14]. These findings suggest that GRs and MRs have a sex-specific influence on the relationship between stress responses in the brain and stress-related mental disorders such as depression. The sex-specific functions and stress reactions of GRs and MRs in the etiology of depression remain relatively unexplored. As such, the roles of GRs and MRs in mental health should be re-examined in a gender-based context. The existence of a genetic risk of GRs and MRs associated with stressful conditions or the recurrence of stress-related mental disorders has been posited. In particular, there is evidence to support the association of various common single nucleotide polymorphisms of GR and MR genes and their derived haplotypes with sensitivity to cortisol, as well as the onset and presence of major depression.

At present, there exist well-researched reviews that have discussed the link between various cortisol sources and stress or depression [15,16,17], or gender-related neuroendocrine changes in depression under specific circumstances [18,19,20]. However, gender differences in cortisol as well as GRs and MRs are topics that would benefit from more coverage, especially given recent clinical findings. In this narrative review, we discuss gender-related specificities that may be present in the link between cortisol release and depression. We will also review the role of GRs and MRs in the HPA axis, and in gender differences in mental disorders from both pre-clinical and clinical studies.

## 2. Structure and Function of Glucocorticoid and Mineralocorticoid Receptors in the HPA Axis

The human glucocorticoid receptor (GR) is coded by the NR3C1 gene, which is located on chromosome 5q31.3, and consists of three main structural and functional domains [11,21]. The human NR3C1 gene consists of 10 exons. Of these exons, the first, exon 1, is an untranslated region (UTR). Meanwhile, the N-terminal modulatory domain (NTD) is encoded by exon 2, while exons 3 and 4 encode the DNA binding domain (DBD). Finally, the hinge region (HR) and ligand-binding domain (LBD) are encoded by exons 5 to 9 [11]. In contrast to other steroid hormone receptors such as progesterone and estrogen, there is no F region to be found in the GR. From the NR3C1 gene, the classic GR isoform as well as the non-ligand-binding GR isoform are produced from the alternative splicing of two terminal exon 9s [22]. The NTD is where transcriptional activation function 1 (AF1) is located and serves as the site for most post-translational modifications [23]. The DBD contains two zinc finger motifs that bind to their target DNA sequences, glucocorticoid responsive elements (GREs) [23]. The LBD is where glucocorticoids bind and contains the transcriptional activation function 2 (AF2) that engages in ligand-dependent interactions with co-regulators [23]. GRs can be found in relatively high quantities throughout the human brain [24], with higher concentrations found in the frontal area, hypothalamus, hippocampus [25], and amygdala [26]. 

The mineralocorticoid receptor (MR) is coded for by the NR3C2 gene, which is located on chromosome 4q31. While the MR is structurally and functionally similar to the GR, differences in the LBD lead to the MR having a higher affinity for ligands such as aldosterone [27,28]. MRs are found throughout the human brain, though they are mainly localized in the limbic region, such as the hippocampus and the dorsolateral septum [29]. 

Since the late 1980s, the roles of glucocorticoids and their receptors, GR and MR, have been widely documented in relation to HPA activity [30]. Upon binding to glucocorticoids, GRs and MRs translocate from the cytoplasm to the nucleus, where they bind to GREs to regulate the transcription of target genes. Heat shock proteins (Hsps) and FK506 binding proteins (FKBP51 and FKBP52) play an important role in this binding and translocation process. Hsp70 inhibits GR ligand binding by partially unfolding it, while Hsp90 reverses that inhibition and promotes the binding of the GR to glucocorticoids [31]. FKBP51 on the other hand maintains the GR in a high ligand affinity state and keeps it in the cytoplasm, while FKBP52 dislodges FKBP51 and promotes the translocation of the GR into the nucleus [32]. Due to the high similarity between the MR and the GR, FKBP52 and Hsp90 have been shown to play the same role in MR nuclear translocation [33].

While GRs and MRs have different binding affinity to glucocorticoids, the high binding affinity of MRs to glucocorticoids, they tend to be saturated by glucocorticoids even at the baseline of HPA activity [30]. The lower relative affinity of the GR means that binding to the GR is only observed during an increase in glucocorticoid circulation, which is seen during periods of stress [30].

There have been more discoveries in recent years about the mechanisms of GRs and MRs in relation to HPA activity. Increased microRNA-124 (miR-124) levels have been associated with major depressive disorder [34]. MiR-124 directly targets GRs and when inhibited alleviates depressive-like symptoms in mice [35]. Furthermore, hypermethylation of NR3C1 exon 1F, which is related to higher basal HPA activity, has been observed in depressed patients who had experienced early life stress [36]. Genomic analyses reveal that differences in haplotypes for N3C1 and N3C2 affects stress-induced reactivity to cortisol, which subsequently impacts cognitive behavior [37]. Chaperone proteins such as Hsps and FKBPs have also been implicated in major depressive disorders, as certain SNPs for Hsps and FKBPs have been linked to a higher incidence of the disease [38,39], while increased FKBP51 expression in particular has been observed in depressed patients [40]. The role of the GR/MR in the HPA axis under depression is summarized in Figure 1.

## 3. Gender Differences of Cortisol in Clinical Studies

### 3.1. Recent Findings in Gender Differences of Cortisol

Recent clinical studies in the PubMed/MEDLINE and EMBASE/Ovid were referenced to examine potential gender differences in cortisol expression. The majority of the studies conducted on salivary samples (Table 1) in the past 10 years discovered no significant correlation between cortisol levels and depression symptoms between gender [8,41,42,43,44,45,46,47,48,49]. However, studies conducted specifically on younger subjects found that cortisol reactivity was higher in male children, particularly during a state of depression [50,51,52,53,54]. Male children tended to have higher cortisol levels than female children in depression [52,53], and while high cortisol levels were a risk factor for both genders, moderate cortisol levels could also be a risk indicator for depression in male children, and accordingly low cortisol levels could be a protective factor only for that gender [54].

With regards to serum cortisol studies (Table 2), two studies reported no correlation between gender and depression in their analysis of serum cortisol samples [55,56], while another study found that there was a correlation between cortisol levels and gender, with males demonstrating significantly higher serum cortisol levels [57]. Furthermore, depressed men also exhibited heightened cortisol reactivity, while depressed women had a blunted cortisol response in comparison [58]. Another study on serum cortisol performed in Japan showed that men had higher cortisol levels than women in the control group but not in the depressed group, whereas depressed women exhibited higher cortisol levels than women in the control group [59].

**Table 1 ijms-24-07129-t001:** Salivary mixed-gender studies investigating the relationship between cortisol and depression.

Reference Number	Country	Type of Study	Duration	Sample Size	Age Group (year)	Type of Cortisol Sample	Main Conclusions
[8]	Nigeria	Cross-sectional	3 months	88	17–25	Saliva	No significant correlation between salivary parameters and depression scores among both genders.
[35]	United Kingdom	Longitudinal	3 days	841	15	Saliva	No difference in association of cortisol awakening pattern and depression according to gender.
[36]	Ecuador	Cross-sectional	4 months	522	11–17	Saliva	No significant association between depression and cortisol according to gender.
[37]	Canada	Longitudinal	2 years	409	3–5	Saliva	Lower cortisol reactivity in female children and high cortisol reactivity in male children showed negative correlation between depressive symptoms and time.
[38]	China	Longitudinal	3 months	85	10–12	Hair, Saliva	Positive correlation among depressive symptoms and hair cortisol seen in males but not females.
[39]	Canada	Cross-sectional	1 week	111	12–18	Saliva	Higher cortisol reactivity seen in depressed males but not females.
[40]	United Kingdom	Longitudinal	3 years	1858	Mean = 13	Saliva	High morning cortisol levels increases risk of depressive disorder among male children with high depression but not females.
[41]	China	Longitudinal	1 year	712	8–11	Saliva	High level of cortisol associated with risk of depressive symptoms in both male and female children, but moderate level of cortisol only shows the same association for males and not females. Low cortisol levels seem to be a protective factor only for male children.
[42]	United States	Longitudinal	3 days	300	16–18	Saliva	Measurement of cortisol awakening response and rhythmicity; no gender differences noted.
[43]	Canada	Longitudinal	2 days	120	20–45	Saliva	Gender does not play a role between both groups. Depression associated with lower morning cortisol.
[44]	United States, Belgium	Continuous-time process model	30 days	621	18–61	Saliva	No relationship between gender, cortisol, and depression. Variability of cortisol associated with gender but not depression. Regulation strength of cortisol associated with depression (specifically chronic depression) but not gender.
[45]	Pakistan	Cross-sectional	8 months	60	>17	Saliva	Mean cortisol levels are not significantly different among depressed males and females.
[46]	United States	Exploratory	1 month	162	12–19	Saliva	No difference in cortisol pattern in depressed males and females.
[47]	United States	Exploratory	2 days	60	Mean = 43.84	Saliva	No significant difference in females and males between depressive symptoms and diurnal cortisol slopes.
[48]	Brazil	Cross-sectional	Unavailable	256	>65	Saliva	No significance in cortisol and depression relation to gender.
[49]	United States	Cross-sectional	2 weeks	196	18–21	Saliva	Lower cortisol level in depressed females but higher cortisol levels seen in depressed males.
[50]	China	Longitudinal	3 days	80	18–45	Saliva, Serum	Measurement of cortisol awakening response and rhythmicity; no gender differences noted.
[51]	United Kingdom	Cross-sectional	1 days	80	20–65	Saliva	No gender differences noted in salivary cortisol.
[52]	Germany	Longitudinal	3 weeks	50	18–65	Saliva	No gender differences noted in salivary cortisol.
[57]	Netherlands	Systematic review and meta-analysis	24 years	1270	Mean = 30.5	Saliva, Serum	Females with depressive disorder had blunted cortisol response while males had increased cortisol response.

**Table 2 ijms-24-07129-t002:** Serum mixed-gender studies investigating the relationship between cortisol and depression.

Reference Number	Country	Type of Study	Duration	Sample Size	Age Group (Year)	Type of Cortisol Sample	Main Conclusions
[53]	Russia	Cohort	2 days	363	18–45	Serum	No significant difference between depressive disorders and cortisol levels between gender.
[54]	Indonesia	Cross-sectional	3 months	79	19–68	Serum	There was a significant correlation between gender and cortisol level with males showing higher cortisol levels.
[55]	United States	Cross-sectional	1 day	131	Mean = 38	Serum	Male patients had significantly higher cortisol levels, but females did not show similar findings.
[56]	Poland	Prospective	6 months	37	Mean = 54.7	Serum	No significant difference between cortisol concentration and depression between females and males.
[57]	Netherlands	Systematic review and meta-analysis	24 years	1270	Mean = 30.5	Saliva, Serum	Females with depressive disorder had blunted cortisol response, while males had increased cortisol response.
[58]	Japan	Cross-sectional	1 day	87	Mean = 50.7	Serum	Males exhibited higher cortisol levels in the control group; females that were depressed exhibited higher cortisol levels than females in the control group.

The studies performed on hair cortisol levels provided conflicting results (Table 3). The Lu study in China [60] reported a positive correlation between hair cortisol and depressive symptoms, though it was observed only in males and not in females. The Berger study [61] on Australian Aboriginal subjects showed no correlation between gender, cortisol, and depression. The Gerber study in Switzerland, on the other hand, reported increased hair cortisol levels correlated with lower self-perceived stress and anxiety, with no gender differences observed [62]. However, these studies were conducted on homogenous samples when it came to race and age group, which may be a reason for this disparity.

Four studies investigated the potential disparities in the circadian rhythmic expression of cortisol and were unable to find any significant differences in the cortisol awakening response and diurnal cortisol slope for men and women suffering from depressive symptoms [8,42,44,48].

### 3.2. Differences in Sampling Methodology

The source of the cortisol samples and the age group sampled appear to have an influence on the results of the studies. Hair cortisol reflects the accumulation of cortisol throughout a period of time, while salivary and serum samples have more fluctuations and do not reflect cumulative patterns [60]. Hence, hair samples would be more suitable for investigations focusing on chronic depression over a longer period of time, where long-term changes in cortisol levels are more important for the study’s purpose. A prior review on hair cortisol found that cortisol concentrations in the hair increase with age, as well as higher hair cortisol concentration in men compared to women in chronic stress studies [17]. This lines up well with one of the papers examined in this review. Both race and age may be a reason for any disparities, as mentioned earlier.

Salivary and serum cortisol samples would be more appropriate sources when the study’s focus is on acute changes in cortisol over the period of a single day, such as investigations into cortisol rhythmicity, or sudden changes in cortisol levels reflecting specific incidents that may have occurred in that day. While a previous systematic review by Knorr et al. [15] also found no gender differences in salivary cortisol for depression, it is possible that cortisol reactivity in early life may exhibit gender differences, as the studies in this review have suggested.

The comparison of males and females of similar ages in these studies suggest a positive correlation between high cortisol levels and depressive symptoms only in younger males, relative to younger females. They also appear to show a higher cortisol reactivity. These differences in gender seem to diminish with age, particularly in salivary sample studies, while any differences in serum cortisol may still exist well into adulthood. For males, this correlation with pubertal age could be linked to the release of androgens during puberty, which reduces the sensitivity of CRH receptors and leads to increased cortisol levels [60]. However, the effect of female hormones on cortisol and depression is yet to be completely understood.

## 4. Gender Differences in Glucocorticoid and Mineralocorticoid Receptors in Mental Health

### 4.1. Gene Expression of Glucocorticoid Receptor- and Mineralocorticoid Receptor-Related Genes

Genetic analysis of GR-related genes revealed that a polymorphism of the NR3C1 GR gene is significantly associated with major depression in women [63]. The frequency of the polymorphism was more than three times higher in women suffering from depression [63]. The same study did not find NR3C2 MR gene polymorphisms to be significantly associated with depression, nor did it discover any male-associated polymorphisms linked to depression [63]. Methylation of the NR3C1 gene also demonstrated similar results; a study conducted in Thailand showed significant methylation levels at the cytosine-phosphate-guanine (CpG)7 group [64]. When examined by gender, methylation levels were vastly different for females but not for males [64]. These results suggest that the methylation and polymorphism of GR genes may be the major driver for depression in females, whereas for males it is the lack of GR expression in itself as indicated earlier.

Examination of specific MR haplotypes showed that haplotype CA is associated with enhanced MR activity and increased behavioral optimism [65]. This haplotype is also linked to a lower risk of depression when examining data from a large genome-wide association study, although the correlations are valid only in women [65]. More studies investigating MR haplotypes discovered other haplotypes that influence susceptibility to depression following childhood maltreatment [66]. A study found that the CA haplotype increased resilience to depression in females [66], but, in males, the CA haplotype increased vulnerability to depression [66]. Conversely, male CG and GA haplotype carriers showed increased resilience, while female GA haplotype carriers demonstrated increased vulnerability [66]. Furthermore, an examination of biomarkers representative of central MR function found that central MR hyperactivity is associated with poor recovery, only in men, after treatment for clinical depression [67].

Besides depression, GRs and MRs have also been associated with other mental health disorders such as schizophrenia and psychosis. Thus far, there has been little to suggest that gender differences in GR expression are present in schizophrenia and other mood disorders. Examination of GR mRNA levels in post-mortem human brain specimens suffering from bipolar disorder and schizophrenia found evidence of GR dysregulation in varying regions of the brain depending on the disorder but no significant gender differences [68]. The same appears to hold true for MR studies. While MR mRNA expression was found to be decreased in the prefrontal cortex of patients suffering from schizophrenia and bipolar disorder, no significant differences were observed in terms of gender [69].

However, more recent research has uncovered gender differences in the methylation of both GRs and MRs in schizophrenia. NR3C1 methylation at exons 1B, 1D, and 1F were found positively correlated with schizophrenia in females, while methylation at exons 1D, 1F, and 1H were found correlated with schizophrenia in males [70]. NR3C2, on the other hand, showed significantly increased methylation of the N3C2-4 region in females with schizophrenia, with no such correlation found in males [71]. Schizophrenia and psychosis are well associated with the methylation of NR3C1 at various stages [72] and any gender differences in methylation for this gene should be an important consideration for future genetic studies on schizophrenia. The clinical findings of GR and MR expression have been summed up in Figure 2.

### 4.2. Gender Differences in Animal Studies of Glucocorticoid and Mineralocorticoid Receptor Expression

The presence of gender differences in GR/MR expression in early-life-stress-induced depression appears to be supported by an early life stress animal model. Reactivity of the HPA axis was reduced in female mice exposed to early life stress and further exacerbated by decreased MR expression [73]. Male mice showed an increase in HPA reactivity as well as elevated upregulation of MR [73]. Gender differences in GR and MR expression are also seen in avian species. Social stress via mate pair separation of zebra finches revealed that female finches displayed an increase in hippocampal MR but not GR, while males showed a decrease in both hippocampal MR and GR [74].

A study performed on socially isolated female rats observed an increase in depressive-like behavior that subsides during estrus along with a reduction in GR expression in the hippocampus [75]. Another experiment performed on juvenile male rats showed GR expression increased and MR expression decreased during social isolation [76]. Furthermore, stress increased GR expression in group-housed rats with time, while it did not affect GR expression in isolated rats [76]. These studies indicate the influence of social isolation on altering GR expression. Furthermore, the finding that ovarian hormones may reduce GR expression [75] suggests a gender difference present in how GR activity is affected by social isolation.

Knocking out GR expression in the forebrain showed different effects on male and female mice [77]. The deletion of GRs increased basal and stress-induced corticosterone in male mice, while no effects were observed in female mice [77]. Male mice also showed elevated depression-like behavior due to GR deletion [77]. GR expression in the hippocampus was shown to significantly increase in depressed elderly women compared to elderly men [78]. This indicates clear gender differences in GR activity of different brain regions during depression. The increased susceptibility of male mice to GR deletion, as well as its relatively reduced expression in elderly male humans, suggests that GR—or its lack thereof—may be more important in the development of depression in males, as compared to females. It is possible that these findings in animal models may serve to inform future clinical studies on GR and MR differences in human males and females.

## 5. Conclusions

There are significant gender differences in the relationship between cortisol and depression. While the results may vary based on the age group and source of the samples, several mixed-gender studies show that there appears to be heightened serum cortisol levels in depressed males compared to depressed females. There are also differences noticed in salivary cortisol reactivity, with male children having increased reactivity, though these differences seem to diminish with age. There have been few studies focusing on male-only effects of cortisol in depression [79,80], while a majority of studies regarding female-only effects have focused on peripartum women [81,82,83,84,85,86]. In this regard, there may be a need for a more varied and diverse collection of data in clinical studies.

Significant gender differences can also be seen in glucocorticoid and mineralocorticoid receptor expression. These differences are found both in expression levels as well as epigenetic regulation depending on the disorder and on the brain region (Figure 2). These clinical findings are also strongly supported by early life stress animal models, which indicate that male mice exhibit increased HPA activity and upregulated MR expression in depression, with female mice showing lowered HPA activity and MR expression. GR deletion also plays a role in male mice depression but not in females.

The interplay of GRs and MRs and mental health is a complicated issue with many factors at play; it would be interesting for future studies to consider gender in their approach to GR and MR research. One possible challenge for the future could be to understand how such sex-specific complicated findings can be reconciled and how GRs and MRs eventually interact and balance each other out in stress responses as well as the onset of depression profiles in males and females. GRs and MRs could become new drug targets either as classic steroid receptors or as specific transcription factors to novel target genes or even nonsteroidal GR-targeting molecules. This will help to answer the ever-greater demand for new clinical approaches for the treatment of stress-related mental disorders.

## Figures and Tables

**Figure 1 ijms-24-07129-f001:**
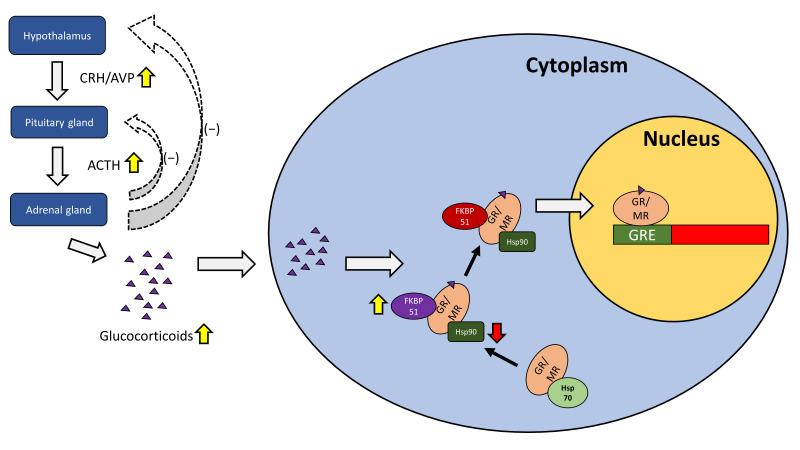
The role of the GR/MR in the HPA axis under depression. The hypothalamus releases corticotropin-releasing hormone (CRH) and arginine-vasopressin (AVP), acting on the pituitary gland which in turn releases adrenocorticotropic hormone (ACTH) to stimulate the adrenal gland. The adrenal gland plays a role in the negative feedback loop for the hypothalamus and the pituitary gland, while releasing glucocorticoids such as cortisol. Glucocorticoids travel into the cell and bind to GRs or MRs, with the help of Hsp90 and FKBP51, before being transported into the nucleus with the aid of FKBP52 where they activate GREs for subsequent gene transcription. During depression, CRH and AVP are elevated, resulting in increased ACTH and increased glucocorticoid release such as cortisol. Chaperone proteins such as FKBP51 may be elevated while others such as Hsp90 may be impaired in function.

**Figure 2 ijms-24-07129-f002:**
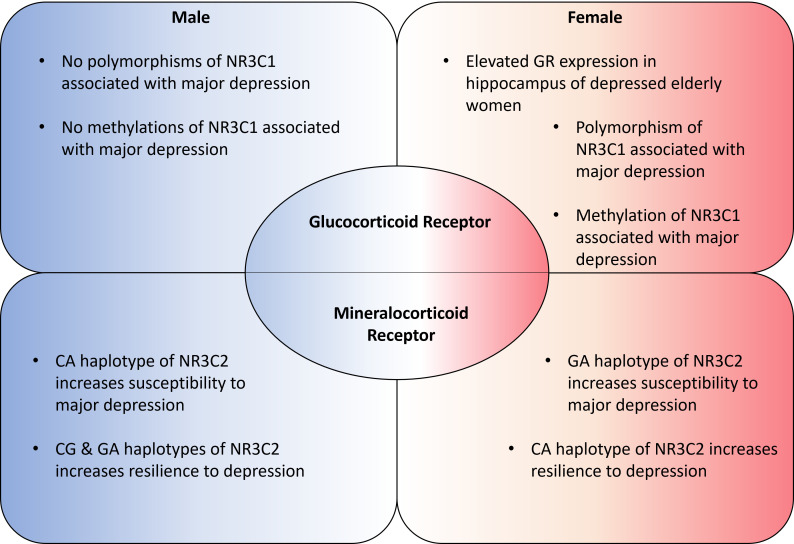
Genetic gender differences in glucocorticoid receptor and mineralocorticoid receptor expressions.

**Table 3 ijms-24-07129-t003:** Hair follicle mixed-gender studies investigating the relationship between cortisol and depression.

Reference Number	Country	Type of Study	Duration	Sample Size	Age Group (Year)	Type of Cortisol Sample	Salient Points
[38]	China	Longitudinal	3 months	85	10–12	Hair, Saliva	Positive correlation among depressive symptoms and hair cortisol only seen in males and not females.
[59]	Australia	Cross-sectional	2–3 months	329	15–24	Hair	No differences in hair cortisol among gender and depressed groups.
[60]	Switzerland	Cross-sectional	7 days	46	Mean = 21.17	Hair	Increased hair cortisol correlated with lower reported self-perceived stress and anxiety; no gender differences observed.

## Data Availability

Not applicable.

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
