# Peer review of "Gender Differences in Cortisol and Cortisol Receptors in Depression: A Narrative Review"

_ijms, 2023, doi:10.3390/ijms24087129_

Round 1
Reviewer 1 Report (Previous Reviewer 2)
Thanks for addressing the reviewer's comment
Author Response
We thank the reviewer for accepting the revisions that have been made.
Reviewer 2 Report (New Reviewer)
This paper is a narrative review of gender differences in cortisol and cortisol receptors with reference to depression.
The association between the hypothalamic-pituitary-adrenal (HPA) axis and depression is a well-documented one, and has been addressed in a large number of translational and clinical studies, the results of which have been summarized in several systematic reviews and meta-analyses. In this context, the authors should clearly specify the novel contribution(s), if any, made by their review to our current state of knowledge in this field. (For example, if they have found that none of the existing systematic reviews or meta-analyses have examined gender differences, this would be a justification for the current review.)
The authors report having retrieved literature from PubMed and Google Scholar. As the latter is likely to retrieve studies of low quality (e.g., from predatory journals) or low relevance, it is not recommended as a stand-alone second source for literature reviews of this sort, despite its easy accessibility and wide coverage. Standard practice for a review of this sort is usually two to three scholarly databases (PubMed, EMBASE, PsycINFO, etc.) which can be supplemented by Google Scholar if desired.
The search strategy has been described briefly in lines 154-166 but requires some clarification. Why were only the last ten years considered for the current review? (If the authors have conceived this review as an update to existing reviews, such as the paper by Knorr et al. 2010, then the ten-year period is justified; however, this should be stated in the text.) It is also advisable to provide a flow diagram showing the process of retrieval and selection of the studies included in the review, and to clearly mention the type of review (scoping, selective, narrative, etc.) in both the title and the Methods section.
The inclusion of some results from animal studies is questionable, as animal models offer only approximations to the syndrome of depression in humans and results from these models cannot be generalized easily. Moreover, findings relevant to gender differences in animals do not apply to humans because of the vast biological and environmental differences involved (e.g., social and cultural factors related to gender roles in humans, which can cause stress and HPA axis dysregulation, and which cannot be replicated in animal models). I would strongly suggest confining the review to studies of human subjects. Animal models can be discussed in the Discussion section if necessary.
Section 3 requires some work in terms of organization. Instead of repeating the findings mentioned in the tables, the authors could summarize the key findings and discuss methodological issues / variations under separate subheadings (3.1, 3.2, etc.)
It would be useful to provide a summary table for the findings pertaining to glucocorticoid receptors in humans, as an addition to Section 4.
A brief Discussion section, summarizing the key findings of this review and contrasting them with the findings of earlier systematic reviews of cortisol in depression, is also required.
In short, this review has identified a large number of relevant studies; however, the way in which these studies are summarized and interpreted requires a substantial amount of revision.
Author Response
We thank the reviewer kindly for their insightful comments. We have uploaded our feedback and revisions in the attachment.

Round 2
Reviewer 2 Report (New Reviewer)
The revisions made by the authors are satisfactory in my opinion. I have no further major changes or corrections to suggest.
This manuscript is a resubmission of an earlier submission. The following is a list of the peer review reports and author responses from that submission.
Round 1
Reviewer 1 Report
In this manuscript, authors compilate a set of clinical studies focused on the relationship of glucocorticoid level found in different types of clinical samples (saliva, serum, and hair follicle) with stress-associated mental disorders, namely depression, and gender differences. In addition, authors comment glucocorticoid receptor (GR) and mineralocorticoid receptor (MR) genetic analysis, as well as data obtained in animal models that support for a gender differential association between GR and MR gene polymorphisms, expression level and/or functionality and mental health. The subject addressed by the authors is relevant for the field and may have consequences supporting for gender specific biomarkers relevant for depression. Nonetheless, I have several concerns on the current manuscript that make me to recommend a major revision before considering acceptance.
1.- In general, authors need to review the whole manuscript to introduce appropriate amendments of many statements that are not correct in the current manuscript. Following there are several examples:
- Lanes 14-15: “Between gender, salivary cortisol showed no correlation between gender”.
- Lanes 90-91: “The human GR is coded by the NR3C1 gene; it is located on chromosome…” should read: “The human GR is coded by the NR3C1 gene, which is located on chromosome…”. The same for the MR on lanes 107 and 108.
- Lanes 99-100: “The NTD is where the transcriptional activation functions 1 and 2 (AF1 and AF2) are located”. AF2 is in the LBD.
-Lane 116: “…where they activate GREs for subsequent gene transcription”. To be correct, GR and MR bind to GREs to regulate transcription of target genes.
- Figure 1. It is not mentioned in the main text.
- The authors use the expression “perinatal women” several times along the manuscript when “peripartum women” would be more correct. The use of “boys” and “girls” should also be avoided.
- The title of figure 2 needs correction, an improved statement could be “GR and MR genetic HPA reactivity differences in males and females”
2.- Data presented in section 4 should also be commented in the abstract.
3.- In relation to Tables:
- They should be mentioned in the main text.
- They should show reference number, country, type of study, duration, sample size, age, type of sample and main conclusions. Other data such as authors, title, year of journal could be found in the reference list. In this way, tables will be handled more easily.
4.- In section 3, the studies performed only in males or in females (current tables S1 and 2) should only be mentioned to support finals conclusions. In this regard, studies in males are too limited (both in number and in group size) to be relevant, and in women the studies are very heterogenous and include pregnant women that have no male counterpart. Therefore, these two tables should be eliminated as well as the part of the main text dedicated to comment all these studies.
This section should be focused in describing clinical studies performed with mixed gender cohorts. Several issues seem to be relevant for the association between glucocorticoids and depression including age, ethnicity, the parameters to define of depression and the type of sample (saliva, serum, and hair follicle), among others. In this regard, table S3 should be formatted as recommended above, and segregated in 3 tables, one for each sample type. The main text should also be rearranged according to this table segregation. This will allow authors to concentrate in saliva samples which are the most numerous and therefore, the results are more robust.
Reviewer 2 Report
The author presented a review manuscript titled "Gender Differences in Cortisol and The Receptors in Depression".
The reviewer had the following comments regarding this manuscript-
1. The manuscript is not clear as to which type of review it belongs to.
2. There is no explicit methodology for the review.
3. Author mentioned (page 3, line 82) "In this review, we will take a comprehensive look at clinical studies from the past". However, he did not examine other databases (only PubMed/MEDLINE).
4. Since many other databases are omitted accuracy of finding needs revision based on the missed database.
